# Characterization of Critical Determinants of ACE2–SARS CoV-2 RBD Interaction

**DOI:** 10.3390/ijms22052268

**Published:** 2021-02-25

**Authors:** Emily E. F. Brown, Reza Rezaei, Taylor R. Jamieson, Jaahnavi Dave, Nikolas T. Martin, Ragunath Singaravelu, Mathieu J. F. Crupi, Stephen Boulton, Sarah Tucker, Jessie Duong, Joanna Poutou, Adrian Pelin, Hamed Yasavoli-Sharahi, Zaid Taha, Rozanne Arulanandam, Abera Surendran, Mina Ghahremani, Bradley Austin, Chantal Matar, Jean-Simon Diallo, John C. Bell, Carolina S. Ilkow, Taha Azad

**Affiliations:** 1Ottawa Hospital Research Institute, Ottawa, ON K1H 8L6, Canada; emibrown@ohri.ca (E.E.F.B.); Reza.Rezaei@uottawa.ca (R.R.); tjamieson@ohri.ca (T.R.J.); jdave@ohri.ca (J.D.); nikmartin@ohri.ca (N.T.M.); rsingaravelu@ohri.ca (R.S.); mcrupi@ohri.ca (M.J.F.C.); sboulton@ohri.ca (S.B.); satucker@ohri.ca (S.T.); jduon069@uottawa.ca (J.D.); joapupo@gmail.com (J.P.); adrian.pelin@turnstonebio.com (A.P.); ztaha@ohri.ca (Z.T.); rarulanandam@ohri.ca (R.A.); absurendran@ohri.ca (A.S.); baustin@ohri.ca (B.A.); jsdiallo@ohri.ca (J.-S.D.); jbell@ohri.ca (J.C.B.); cilkow@ohri.ca (C.S.I.); 2Department of Biochemistry, Microbiology and Immunology, University of Ottawa, Ottawa, ON K1H 8M5, Canada; 3Department of Cellular and Molecular Medicine, Faculty of Medicine, University of Ottawa, Ottawa, ON K1N 6N5, Canada; hyasa068@uottawa.ca (H.Y.-S.); Chantal.Matar@uottawa.ca (C.M.); 4Department of Biology, University of Ottawa, Ottawa, ON K1N 6N5, Canada; mina.ghahremani88@gmail.com

**Keywords:** SARS-CoV-2, angiotensin-converting enzyme 2, receptor binding domain, NanoLuc Binary Technology, spike protein, vaccine development, drug development, bioluminescence

## Abstract

Despite sequence similarity to SARS-CoV-1, SARS-CoV-2 has demonstrated greater widespread virulence and unique challenges to researchers aiming to study its pathogenicity in humans. The interaction of the viral receptor binding domain (RBD) with its main host cell receptor, angiotensin-converting enzyme 2 (ACE2), has emerged as a critical focal point for the development of anti-viral therapeutics and vaccines. In this study, we selectively identify and characterize the impact of mutating certain amino acid residues in the RBD of SARS-CoV-2 and in ACE2, by utilizing our recently developed NanoBiT technology-based biosensor as well as pseudotyped-virus infectivity assays. Specifically, we examine the mutational effects on RBD-ACE2 binding ability, efficacy of competitive inhibitors, as well as neutralizing antibody activity. We also look at the implications the mutations may have on virus transmissibility, host susceptibility, and the virus transmission path to humans. These critical determinants of virus–host interactions may provide more effective targets for ongoing vaccines, drug development, and potentially pave the way for determining the genetic variation underlying disease severity.

## 1. Introduction

The severe acute respiratory syndrome coronavirus 2 (SARS-CoV-2) is the etiologic agent responsible for the COVID-19 pandemic and is an ongoing worldwide public health threat. As of 13 February 2021, there have been over 100 million confirmed cases, and over 2 million confirmed deaths resulting from SARS-CoV-2 infection (WHO reports, https://www.who.int/emergencies/diseases/novel-coronavirus-2019/situation-reports (accessed on 1 February 2021). Tremendous efforts are currently underway to develop rapid drug screening methodologies and novel vaccines. The large variability of disease severity among individuals infected with SARS-CoV-2 continues to be investigated and new findings in this field may shed light on strategies to tailor these new therapeutics to patients [1].

Entry of SARS-CoV-2 is mediated by interaction of the viral Spike glycoprotein (S) with its main target receptor angiotensin converting enzyme 2 (ACE2), found on the surface of mammalian cells, primarily in the lower respiratory tract [2]. The S protein exists on the viral surface as a trimer and is composed of two subunits, S1 and S2, which cooperatively play a role in viral entry and fusion of the viral membrane with host cell membrane [3]. Binding to ACE2 is mediated by the receptor binding domain (RBD), located in the C-terminus of the S1 subunit [4]. The identification of amino acid residues that are crucial for the interaction of RBD with ACE2 is of great interest to gain a better understanding of the interplay between viral entry and host genetic factors which may contribute to the observed variability in disease pathogenesis. As such, characterizing functional mutants of RBD and ACE2 may provide critical insights for the development of drugs and vaccines.

The development of biosensor technology is a highly valuable and sensitive analytical tool with a broad spectrum of applications, such as diagnosis and drug development [5,6]. Biosensors designed to emit bioluminescence often rely on luciferase, a class of enzymes that catalyze substrate to produce a bioluminescent signal. One such tool is the NanoLuc Binary Technology (NanoBiT), which enables rapid analysis of protein–protein interactions through use of Nanoluciferase, a small luciferase reporter [6,7,8,9,10,11,12,13]. By exploiting this technology, we have recently developed an assay to rapidly investigate RBD and ACE2 interactions. In addition, this assay can be used to elucidate the impact of both RBD and ACE2 amino acid mutations on their binding abilities, as well as their potential implications for drug development and evaluating immune responses.

Previous studies have examined crucial residues in ACE2 and the SARS-CoV-1 Spike domain [14,15]. Similarly, recent studies have begun to unravel important interactions with SARS-CoV-2 RBD and ACE2 [16]. While some progress has been made towards examining the impact of specific mutations within the SARS-CoV-2 RBD and ACE2, there remains much to be studied with regards to their impact on binding, infectivity, and host susceptibility to viral infection. In addition, the impact of specific RBD mutations on the efficacy of potential therapeutics has not been explored. For example, most vaccines under development are designed to contain the RBD domain of S, and sequencing data has shown that RBD is among the most non-conserved domain in SARS-CoV-2 S [17]. Thus, the question that still remains is whether SARS-CoV-2 viruses harboring select RBD mutations would be controlled by immune responses mounted against the RBD sequence encoded in the original virus strain. Another unexplored area is whether variations in host ACE2 sequences could alter virus susceptibility and/or disease severity among individuals or species. Comparisons on how mutations in ACE2 affect binding of the RBD from SARS-CoV-1 versus SARS-CoV-2 have yet to be performed. We believe that elucidating which variations may have an impact in the ACE2: SARS-CoV-2 S binding affinity is worth investigating since this information could impact the development of therapeutic options for COVID-19 patients. Additionally, we investigate whether any of these critical amino acid sites in ACE2 exist in the human population and may explain severity of the disease.

## 2. Results and Discussion

The first crucial step of SARS-CoV-2 viral entry is mediated by binding of RBD to ACE2, its main cognate receptor, expressed on the surface of the human airway epithelium (Figure 1A). In this study, we aim to investigate whether selected mutations in both SARS-CoV-2 S protein RBD and its host receptor ACE2 could impact their interactions with one another. We also examine how specific mutations in ACE2 and RBD may alter the efficacy of drugs and neutralizing antibodies being developed for treatment and disease prevention purposes, respectively. To accomplish these aims, we use a NanoBiT SARS-CoV-2-RBD and ACE2 biosensor, previously developed by our lab, to initially characterize the RBD: ACE2 interaction (Figure 1B) [18]. The molecular basis of this technology involves the fusion of a Large Bit (LgBiT) subunit to one of the proteins of interest, and the fusion of a Small Bit subunit (SmBiT) to the second protein being investigated. As illustrated in Figure 1B, LgBiT and SmBiT alone have poor affinity for one another; however, both subunits interact to produce a bioluminescent signal in the presence of furimazine, the substrate for Nanoluciferase, if the fused target proteins interact with each other.

### 2.1. Mutations within the ACE2 Ectodomain Alter Binding Affinity of ACE2 with SARS-CoV-2 RBD, SARS-CoV-2 S1 Subunit, and SARS-CoV-1 RBD

ACE2 is a key interacting partner involved in SARS-CoV-2 viral entry, thus we first performed in silico mutagenesis analysis to assess putative ACE2 mutants that could be potentially defective at interacting with SARS-CoV-2 RBD. Based on the overall 3D crystal structure analysis of ACE2 bound to RBD (Figure 2A,B), we selectively identified 22 sites in ACE2, which are most likely involved in ACE: RBD direct interaction (Figure 2A–D). To analyze the contribution of these 22 mutants on their ability to interact with RBD using our NanoBiT based biosensor technology, we first engineered the 22 amino acid mutants of ACE2 (Figure 2E) and linked them to SmBiT. Expression of ACE2 mutants from transfected HEK293T cells was demonstrated by immunoblotting (Figure 3A).

We then investigated the binding affinity of ACE2 wild-type and mutants with both SARS-CoV-2-RBD and SARS-CoV-2 S1 as complementary binding partner. The rationale for also including an S1-based NanoBiT binding partner in our assays was that RBD is encompassed within the S1 subunit of the S glycoprotein. Hence, including S1 would more closely mimic SARS-CoV-2 S behavior in the context of viral infection. Thus, we proceeded to compare whether S1 fused to LgBiT and RBD-LgBiT constructs have similar binding affinity for ACE and its mutants. When combined with SARS-CoV-2-RBD-LgBiT, 12 of the 22 mutants of ACE2 showed a significant decrease in their binding affinity (Figure 3B). Specifically, ACE2 mutants Q24A, F28A, D38A, Y41A, K353A, G354D, D355A, R357A, and NFS (residues 82–84) demonstrated reduced binding to SARS-CoV-2-RBD (Figure 3B). We also found that 13 of the 22 mutants showed a significant decrease in binding to S1 (Figure 3C). Specifically, ACE2 mutants Q24A, F28A, D38A, Y41A, Q42A, L45A, M82A, Y83A, K353A, G354D, D355A, R357A, and NFS had reduced binding to SARS-CoV-2 S1 (Figure 3C). In summary, we found that the same 12 ACE2 mutants that showed decreased binding to SARS CoV-2 RBD also has impaired interactions with SARS CoV-2 S1, with the addition of mutant Y83A that reported a reduced affinity for SARS-CoV-1 S1 only.

Since structural analysis has shown that certain residues in SARS-CoV-2 RBD are well conserved within SARS-CoV-1 RBD [18], we decided to also combine the various ACE2-SmBiT mutants with a LgBiT-SARS-CoV-1 RBD construct. This allowed us to also evaluate how ACE2 mutations could impact ACE2: SARS-CoV-1 RBD interaction. In Figure 3D, we further show that 14 different ACE2 mutations (T27D, F28A, H34A, E35A, E37A, Y41A, Q42A, L45A, Y83A, K353A, G354D, D355A, R357A, and NFS) significantly alter its respective binding efficiency to the wild-type SARS-CoV-1 RBD. Of note, ten of these ACE2 mutants are common between SARS-CoV-1 RBD and SARS-CoV-2 RBD binding assays. This data suggests that in comparison with SARS-CoV-1 RBD, SARS-CoV-2 RBD may be slightly more resistant to ACE2 mutants, which may arise from single nucleotide polymorphisms (SNPs). This observation may have implications with regards to viral susceptibility of various species or individuals based on their ACE2 gene sequence.

### 2.2. SARS-CoV-2 Spike Pseudotyped Lentiviral Particle Assay Confirm That Mutations within the ACE2 Ectodomain Alter Binding Affinity of ACE2 with SARS-CoV-2 RBD

To reinforce the ACE2 mutational findings observed with our biosensor technology described above, we then decided to utilize a lentiviral-based pseudovirus infectivity assay. We hypothesized that ACE2 mutants which retain their binding capacity to SARS-CoV-2 RBD should act as competitive inhibitors for SARS-CoV-2 pseudovirus binding to ACE2 in the host cell, and thus reducing its infectivity (Figure 4A). To test this notion, we combined each of the 22 ACE2 mutants with SARS-CoV-2 spike pseudotyped lentivirus expressing luciferase, and then added the pre-treated pseudovirus samples to HEK293X cells expressing ACE2 and TMPRSS2 (the host protease responsible for cleaving S glycoprotein [19]). We then measured the infectivity with a luciferase assay—a lower signal is indicative of reduced infection, resulting from ACE2 mutants retaining RBD binding capacity, thereby acting as competitive inhibitors (Figure 4B). These results very closely align with mutant binding affinity changes observed with our biosensor technology (Figure 3B,C).

### 2.3. Utilizing ACE2 Sequences as a Predictor for the SARS-CoV-2 Susceptibility of Various Species and Mutation Prevalence in Humans

Our findings suggest that identification of ACE2 amino acid sites which impact RBD binding may provide insights as to SARS-CoV-2 susceptibilities of different individuals or species based on their genetics. Using multiple sequence alignments of the 12 target mutation sites identified to be essential for RBD binding (Figure 3), we propose a method to predict SARS-CoV-2 virus susceptibility (Figure 5A). It is plausible to hypothesize that any species with an identical ACE2 amino acid sequence to the human ACE2 receptor would theoretically be highly susceptible to the virus, and possibly act as a source of transmission, such as the common chimpanzee (*Pan troglodytes*; Figure 5A). With the presence of only one or two differences at key ACE2 binding sites, it is likely that species such as *Tursiops truncatus* (bottlenose dolphin) or *Cricetulus griseus* (Chinese hamster) would still be infected by SARS-CoV-2. In contrast, species such as domesticated horses (*Equus caballus*) or brown rats (*Rattus norvegicus*), which have many differences at key ACE2 residues, are expected to be less susceptible to SARS-CoV-2 infection, due to reduced binding efficiency between ACE2 and RBD. With reduced binding efficacy, viral infection and spread would likely be impaired in these animals, leading them to pose less of a risk for shedding large amounts of infectious virus potentially leading to reduced viral pathogenesis (Figure 5A). These findings are consistent with literature studies of optimal SARS-CoV-2 disease models [20]. For example, increasing studies have indicated that hamsters are a better suited animal model to study SARS-CoV-2 in comparison to mice [21].

Looking at the prevalence of these mutations in people, we observe that some of these SNPs (namely NFS, G354, and D355) already exist in a small fraction of the population (Figure 5B). Our data suggest the possibility that individuals harboring these SNPs may be more resistant to the SARS-CoV-2 virus infection. Genetic differences in the ACE2 receptor of individuals may also provide a partial explanation for the variability in disease severity among individuals. Interestingly, in silico studies from other groups have also suggested that genetic influences cause interindividual variability in ACE2 expression [22,23,24]. In this study, we experimentally validated for the first time the effects of ACE2 mutants on RBD binding affinity, which may pave the path for clinical studies.

### 2.4. Mutations in SARS-CoV-2 RBD That Alter Its Ability to Bind to ACE2 Do Not Impact the Activity of a Competitive Inhibitor nor the Efficacy of a Neutralizing Antibody

A key concern surrounding RNA viruses, and viruses in general, is their capacity to evolve through the accumulation of mutations. Such mutations may drive resistance to therapeutics and/or evasion of host immune responses acquired through infection or vaccination against the wild-type virus. Recently two mutations in the SARS-CoV-2 RBD have been reported in Africa and the UK. We have investigated one of the critical mutation sites reported by the UK, asparagine residue at location 501 (N501), by altering the residue to both to alanine and tyrosine in silico (Figure 6A–D). The change to alanine preserved the internal interactions of the residue with two glutamine residues at positions 498 and 506 but abolished the interaction with tyrosine 41 in ACE2. In contrast, mutation from asparagine to tyrosine preserved all internal interactions and slightly strengthened them. Moreover, this mutation introduced a new strong interaction site, 2.3 A, with lysine 353 in the ACE2 structure. Another recently discovered mutation is the change of glutamic acid at position 484 to lysine. We did not detect any significant change in internal or RBD-ACE2 interaction sites which may be due to the fact that this residue is located in a very flexible loop structure (Figure 6E,F).

Additionally, we selected 16 of the 25 RBD mutants used in our recent study which retained some level of ACE2 binding capacity to further assess the efficacy of a competitive inhibitor and a neutralizing antibody with our biosensor assay. From a therapeutic perspective, recombinant RBD is under clinical consideration for use as a potential viral attachment inhibitor, given its ability to competitively inhibit SARS-CoV-2 virus particles [25]. We found that recombinant RBD expressed from mammalian cells was capable of out-competing all 16 of the RBD mutants tested (Figure 6I). These results suggest that competitive drugs developed to combat the wild-type virus are still applicable to virus strains which may acquire these mutations in the RBD domain leading to reduced receptor binding over time. Neutralizing antibodies (nAb) play a key role in generating protective immunity against viral agents, and are developed through exposure to the virus itself, or to viral antigens in the form of a vaccine. We found that addition of a nAb significantly decreased, if not abolished, the interaction between all examined RBD mutants and ACE2 (Figure 6J). Importantly, this data may indicate that these receptor binding mutations do not provide an immune evasive advantage to the virus, and as a result, should not pose a serious threat to the efficacy of developed vaccines, and immunity to viruses harboring these mutations.

### 2.5. Conservation of RBD across Coronavirus Strains

We used multiple sequence alignment tools to further investigate amino acid conservation patterns of the RBD mutants under examination across multiple betacoronavirus strains. This analysis could provide interesting insights on the possible contributions of these specific residues in RBD and their link to infectivity. The variation in amino acid sites is indicative of which amino acids are critical for maintaining the structural integrity of RBD, and which can be modified for therapeutic purposes. For example, amino acid C361 is present in the bottom of the RBD core, and not at the contact site. Considering the cross-species conservation of this residue, it may exert an important role in stabilizing the overall structure of the protein. Of the analyzed amino acids, K417 was the only one which was not part of RBM yet is still in close proximity to the contact site, providing important contact with residue D30 in the ACE2 structure [4]. Of the other betacoronavirus species analyzed in this study, the presence of K417 is limited to SARS-CoV-2. This unique event could be one of the possible reasons for which SARS-CoV-2 exhibits a higher binding efficiency to ACE2 compared to other viruses in the same family. It is also likely to expect that betacoronaviruses harboring low amino acid sequence similarity to SARS-CoV-2 likely utilize a different host receptor for viral entry. In addition, when analyzing tyrosine residues at positionY449 and Y505, it seems that the hydroxyl (OH) group in the tyrosine side chain may possibly contact two different amino acids on ACE2 structure, appearing critical for effective binding of the receptor-ligand structure. Precisely, Y449 contacts D38 and Q42 in ACE2, and Y505 contacts E37 and R393 in ACE2, respectively [4].

Elucidating which amino acids are conserved and occupy critical roles in viral ligand-host receptor binding could deepen our understanding of antiviral drug development. To increase the spectrum of antiviral agents directed towards betacoronaviruses, targeting conserved amino acids which are crucial for receptor binding may prove to be an interesting strategy. Given the conservation of amino acids between SARS-CoV-2, SARS-CoV-1, and MERS, it would be of interest to develop therapeutics targeting this triad of viruses.

It has been theorized that the SARS-CoV-2 virus originated from bats, was then transmitted to pangolins, and finally spilled over to humans [26,27,28]. We examined the conservation of the 25 RBD point mutants generated in our study to determine if variations in amino acid sequence would follow the predicted path of interspecies transmission. Interestingly, these sites were mostly conserved across all three species, with only three sites (V445, F486, and Y505) differing in bat CoV, when compared to pangolin Co-V and human SARS-CoV-2, and only one site (K417) differing in pangolin CoV compared to the bat and human viruses (Figure 7). These observations are in line with a recent report by Liu et al. [28]. In this study, the authors aligned the full-length spike sequence of SARS-CoV-2, Pangolin-CoV and Bat-CoV-RaTG13. These results and our predictions reported herein support the hypothesis that the path of viral transmission originated from bats, continued to pangolins as intermediate hosts, and then followed its dispersal route to the human population.

## 3. Materials and Methods

### 3.1. Cell Lines

HEK293T (ATCC^®^ CRL-3216) human embryonic kidney cells were obtained from ATCC. Cells were maintained in Dulbecco’s Modified Eagle’s Medium (Gibco, Gaithersburg, MD, USA), supplemented with 10% FBS (VWR, Mississauga, ON, Canada) and 1% penicillin/streptomycin (Invitrogen, Grand Island, NY, USA). Cells were incubated in a humidified 37 °C incubator at 5% CO_2_. Cells are routinely tested for mycoplasma by PCR testing and used for up to 20 passages after thawing.

### 3.2. Plasmids

Engineered inserts are outlined in Appendix A (GenScript, Piscataway, NJ, USA). All biosensor subunits were cloned into the BamHI/NotI sites of pcDNA3.1 to generate mammalian expression constructs.

### 3.3. Transient Transfection

HEK293T cells were grown in 100 mm or 150 mm cell culture dishes to 70% confluence. Cells were transfected with 10 µg of DNA using PolyJet (SignaGen, Frederick, MD, USA) according to manufacturer’s protocol. Forty-eight hours post-transfection, cell supernatants or cell lysates were harvested for subsequent testing.

### 3.4. In Vitro NanoBit Assay

293T cells were lysed using passive lysis buffer (Promega, Madison, WI, USA). NanoBiT assays were performed using native coelenterazine (CTZ; 3.33 mM final concentration; Nanolight Technologies—Prolume Ltd., Pinetop, AZ, USA). Synergy Microplate Reader (BioTek, Winooski, VT, USA) was used to measure luminescence. Results are presented as raw RLU (Relative Luminescence Unit) or normalized to control where indicated. The data presented are the mean of three biological replicates.

### 3.5. Western Immunoblotting

293T cells were lysed using passive lysis buffer. Protein concentration in clarified whole cell lysates was quantified using bicinchoninic acid assay (BCA) (Pierce, Clinton, NY, USA). Then, 20 µg of total protein was prepared in 1× Laemmli buffer, loaded into 4–12% gradient Bis-Tris acrylamide gels and resolved. Following transfer onto nitrocellulose membrane, blots were incubated in 0.1% Ponceau S solution (in 5% *v/v* acetic acid-distilled water) for 5 min at room temperature, imaged, then washed with TBST for 10 min. Blots were then blocked using 5% skimmed milk powder in TBST for 1 h. Blots were subsequently probed using anti-beta-actin primary antibody (1:5000), anti-HA tag (1:1000), or anti-FLAG tag (1:1000) in 5% milk overnight at 4 °C. Following washing with TBST, membranes were probed with HRP-conjugated secondary antibodies (1:10,000) in 5% milk at room temperature for 1 h. Blots were washed with TBST, then developed using BioRad Clarity-Western ECL and the BioRad ChemiDoc imaging system.

### 3.6. Lentiviral Pseudovirus Assay

Plasmid encoding SARS-CoV-2 spike pseudotyped lentivirus was kindly provided by Dr. Jesse Bloom (Fred Hutchinson Cancer Research Center, Seattle, WA, USA) and lentivirus was generated as previously described [29] HEK293T cells were seeded in 6 wells plates to 70% confluency and were transfected with 1ug of individual smBiT ACE2 mutant DNA using PolyJet (SignaGen, Frederick, MD, USA) according to manufacturer’s protocol. 48 h post-transfection, supernatants of transfected cells were harvested. Pseudotyped lentiviruses were incubated with each of the smBiT-ACE2 mutant containing supernatants for 1 h at 37 °C. Immediately following the incubation period, polyclonal HEK293X cells stably overexpressing ACE2 and TMPRSS2 were transduced in 96 well format (300,000 cells). At 4 h post-transduction, all media was replaced. At 48 h post-transduction, cells were lysed using 1× PLB; cell lysates were transferred to an opaque 96-well plate, and infection efficiency was measured by luciferase assay using the Bright-Glo Luciferase Assay system following manufacturer’s protocols (Promega, Madison, WI, USA).

### 3.7. Statistical Analysis

All graphs and statistical analyses were generated in Excel or GraphPad Prism v.8. Means of two groups were compared using two-tailed unpaired Student’s *t*-test. Means of more than two groups were compared by one-way ANOVA with Dunnett’s or Tukey’s multiple comparisons correction. Alpha levels for all tests were 0.05, with a 95% confidence interval. Error is graphed as the standard deviation (SD). Measurements were taken from distinct samples (biological replicates). For all analyses, * *p* < 0.05, ** *p* < 0.01, *** *p* < 0.005, n.s. = not significant.

### 3.8. Multiple Sequence Alignment of ACE2 and Spike Sequences, and Finding ACE2 SNPs in the Target Positions

All of the represented sequences were retrieved from NCBI refseq protein database. Multiple sequence alignment (MSA) was performed using Clustal Omega [30] algorithm with default parameters. Alignments were visualized and prepared for illustration using Jalview software [31]. The multiple sequence alignment of ACE2 polypeptides was reordered based on the binding efficiency data published by another group [32]. The NCBI dbSNP database was searched for possible single nucleotide polymorphisms (SNPs) in target mutation sites and the related information were gathered in a table using Microsoft Excel.

## 4. Conclusions

In this study, we have been the first to provide a comprehensive overview of the effects of select mutations in both the ectodomain of ACE2 and the RBD of SARS-CoV-2. Our findings suggest that ACE2 SNPs in the human population may account for the variability in infectivity and disease progression in the context of SARS-CoV-2 infection. Furthermore, our observations from RBD mutational scanning also provide potential sites for drug targeting and therapeutic development. By combining ACE2 and RBD mutational analyses, we provide insights into the genetics underlying virus susceptibility and the development of possible therapeutic targets.

## Figures and Tables

**Figure 1 ijms-22-02268-f001:**
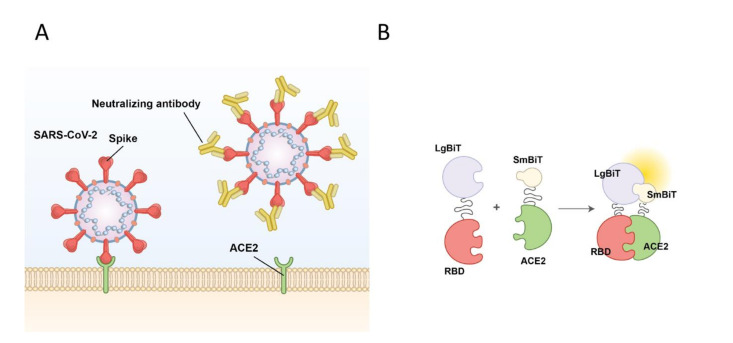
Schematic of SARS-CoV-2 viral entry and the NanoBiT biosensor. (**A**) Schematic of SARS-CoV-2 viral entry via the interaction between the receptor binding domain (RBD) of the spike glycoprotein and the host cell receptor angiotensin-converting enzyme 2 (ACE2). (**B**) Illustration of the NanoBiT complementation-based biosensor which detects interactions between RBD fused to Large Bit (LgBiT) and ACE2 fused to Small Bit subunit (SmBiT).

**Figure 2 ijms-22-02268-f002:**
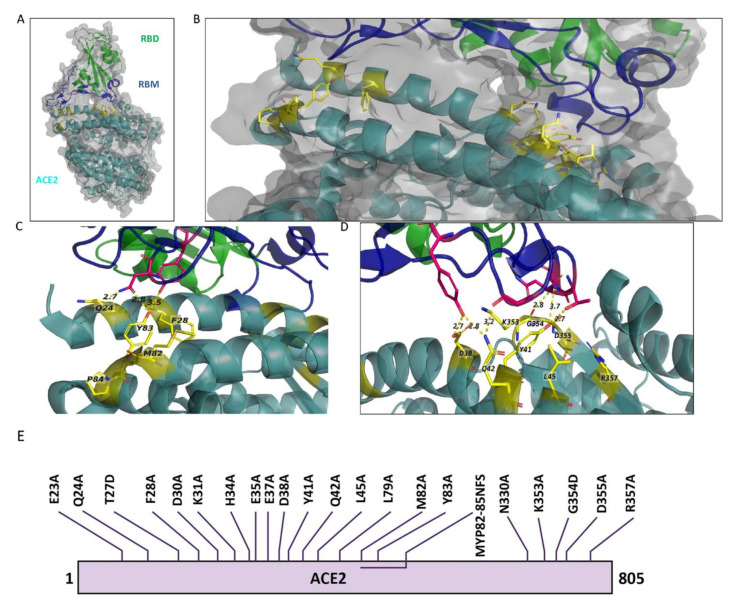
ACE2 amino acids in 3D structure of the bound RBD and ACE2 and schematic representation of mutations in ACE2. (**A**) 3D illustration of the overall structure of RBD bound to ACE2. RBD is colored in green, receptor-binding motif (RBM) of RBD is in dark blue, and ACE2 is in dark cyan. (**B**) Enlarged view of the overall structure depicting ACE2 target mutation sites in stick rendition. Sticks are represented in yellow color. (**C**,**D**) ACE2 target mutation sites in stick representation at the contact site of two molecules. Dotted lines connect the mutant AAs to their contacting AAs in RBM. The structure is from PDB: 6M0J based on the information provided in Lan et al., 2020. (**E**) Illustration of the amino acid changes used to examine ACE2 in this study.

**Figure 3 ijms-22-02268-f003:**
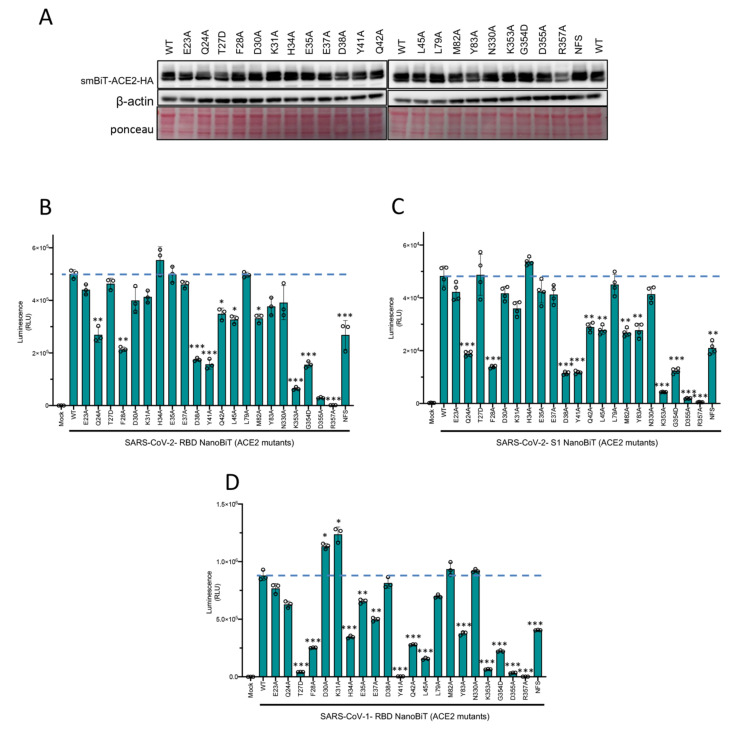
ACE2 mutational analyses with CoV-NanoBiT facilitates delineation of critical host and viral determinants of ACE2–RBD interaction. (**A**) Immunoblot of SmBiT-ACE2 mutant expression from the cell lysates of transfected HEK293T cells. β-actin and total protein loading are shown as controls. (**B**–**D**) Biosensor assay with SmBiT-ACE2 mutants and (**B**) LgBiT-SARS CoV-2 RBD, (**C**) LgBiT-SARS CoV-2 Spike S1, or (**D**) LgBiT- SARS-CoV-1 RBD demonstrating altered binding affinity of various mutants. * *p* < 0.05, ** *p* < 0.01, *** *p* < 0.005.

**Figure 4 ijms-22-02268-f004:**
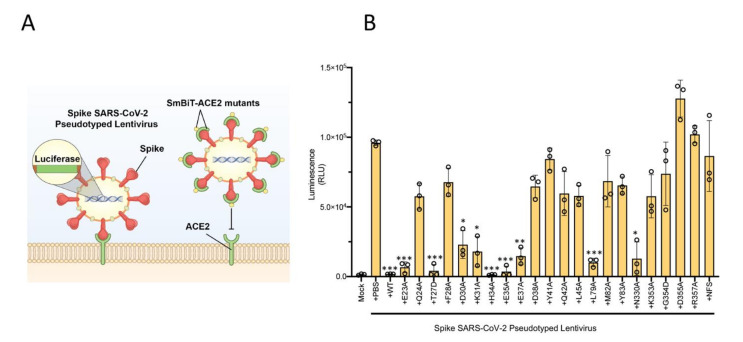
Lentiviral pseudovirus infectivity assay of ACE2 mutational analyses reinforces critical host and viral determinants of ACE2–RBD interaction. (**A**) Schematic of the lentiviral SARS-CoV-2 pseudovirus infectivity assay demonstrating spike-pseudotyped lentiviral attachment to ACE2 host receptor or mutant ACE2 proteins. (**B**) Lentiviral pseudovirus infectivity assay demonstrating the capacity of each mutant examined to act as a competitive inhibitor for the spike–host ACE2 interaction. * *p* < 0.05, ** *p* < 0.01, *** *p* < 0.005.

**Figure 5 ijms-22-02268-f005:**
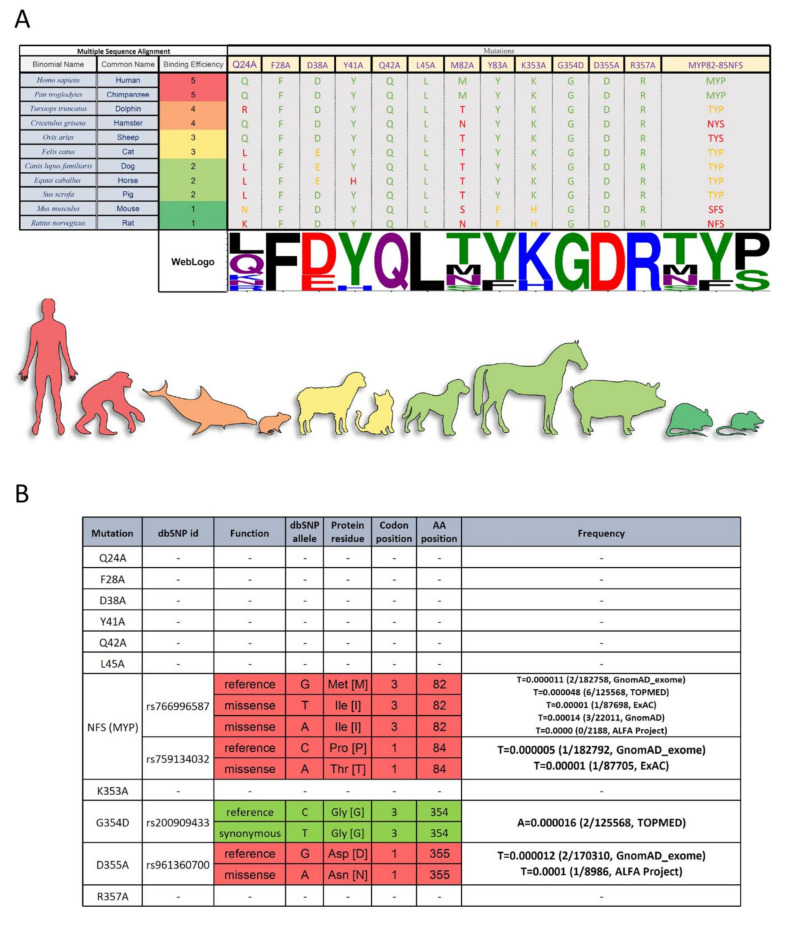
Multiple Sequence Alignment Analysis and single nucleotide polymorphisms (SNP) Frequency Analysis of ACE2 Mutants. (**A**) Sequences were reordered based on their binding efficiency to RBD. Only the target mutation sites are shown, and the non-target amino acids (AA) has been cut out from the alignment representation. The alignment coloring scheme is based on the chemical properties of each AA (each Human AA is green, the AA from the same chemistry is orange, and AA with different chemistry is red. Except for the last column with three AAs which the number of identical AAs to the human determines the color. WebLogo (http://weblogo.threeplusone.com (accessed on 1 February 2021)) representation shows a relative scale of the presence of each AA in each column. (**B**) dbSNP data on single nucleotide Polymorphism in the ACE2 nucleotide sequence on the target amino acid codons. The frequency of each SNP in different datasets was gathered in the last column. A synonymous mutation (no effect on the protein sequence) is green and missense mutations are red.

**Figure 6 ijms-22-02268-f006:**
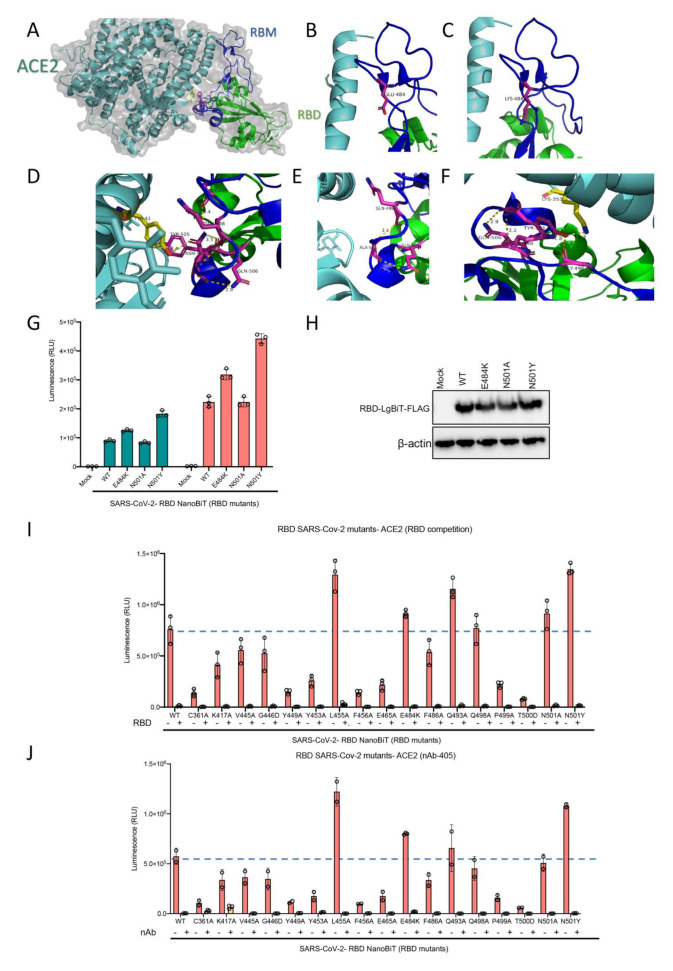
(**A**–**F**) Visualization of RBD N501 and E484 mutations. (**A**) Overall structure of RBD and ACE2 interaction. (**B**) The loop structure in RBD containing E484 residue. (**C**) Predicted structure following E484 residue mutation to lysine which would not significantly change the nature of its interaction in the loop structure of RBD, (**D**) The N501 residue and its interacting residues in RBM and ACE2. (**E**) Predicted structure when the N501 residue mutates to alanine which would eliminate ACE2 interaction but preserve internal RBM interactions. (**F**) Predicted structure when the N501 residue mutates to tyrosine which would preserve and slightly boost internal interactions, and introduce a stronger interaction site with ACE2 compared to the wild type residue. (**G**) ACE2 demonstrating altered binding affinity of various mutants of RBD-LgBiT-mutants. (**H**) Immunoblot of RBD-LgBiT mutant expression from the cell lysates of transfected HEK293T cells. β-actin is shown as control. Competition biosensor assay with LgBiT-RBD mutants which maintained any binding capacity pre-incubated with (**I**) recombinant RBD protein or (**J**) neutralizing antibody to examine therapeutic implications of various mutants.

**Figure 7 ijms-22-02268-f007:**
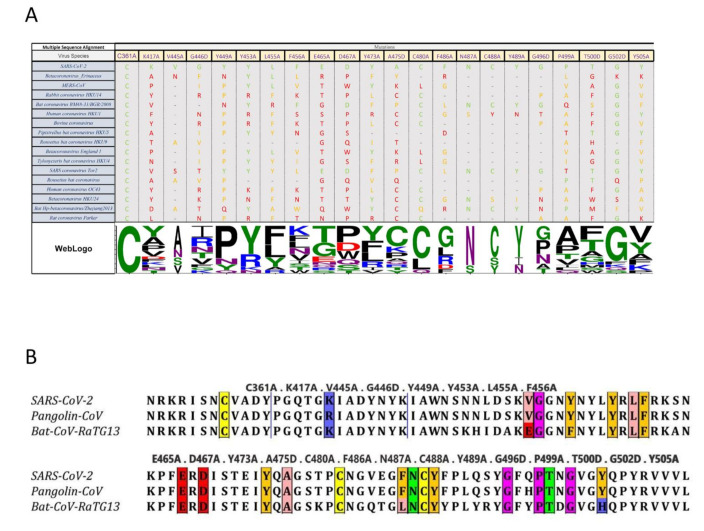
Multiple Sequence Alignment Analysis of RBD Mutants. (**A**) Only the target mutation sites are shown, and the non-target amino acids (AA) has been cut out from the alignment representation. The alignment coloring scheme is based on the chemical properties of each AA (each SARS-CoV-2 AA is green, the AA from the same chemistry is orange, and AA with different chemistry is red. WebLogo (http://weblogo.threeplusone.com (accessed on 1 February 2021)) representation shows a relative scale of the presence of each AA in each column. (**B**) Cross species comparison of RBD amino acids in SARS-CoV-2, Pangolin-CoV 2020, and Bat-CoV-RaTG13. Selected mutants are identified by colored boxes. Colored changes indicate RBD AA differences (V445A and F486A and Y505A).

## Data Availability

Data available on request.

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
