# Peer review of "Characterization of Critical Determinants of ACE2–SARS CoV-2 RBD Interaction"

_ijms, 2021, doi:10.3390/ijms22052268_

Round 1
Reviewer 1 Report
The paper is well written and designed.
The experiments were performed in a rigosoud manner.
In my opinion the present paper is suitble for publication in IJMS.
Author Response
We appreciate the recognition of the impact of our work by the reviewer.
Reviewer 2 Report
The article contains important, interesting, and extremely relevant information about the features of the receptor-binding domain (RBD) interaction of SARS-CoV-2 Spike glycoprotein and one of its targets, angiotensin-converting enzyme 2 (ACE2). The work combines well virtual methods that allow evaluating such interactions, with experimental approaches that confirm the assumptions made. The work is large and versatile. In my opinion, one could even break it down into several articles, in which one could focus on the individual aspects touched upon. For example, separately highlight the evolutionary aspects of the studied interaction. However, of course, it is possible to publish the article in the presented form without further revision after eliminating the following remarks.
- First of all, it is necessary to completely rewrite the abstract, removing obvious statements, and clearly and concisely present what was done in the work. Thus, the mention of the “addition of competitive inhibitors” leads the reader to assume that some special compounds have been investigated, although we are talking about protein fragments of the spike protein. So it should be written. Mention may be made of the lentivirus approach, evolutionary aspects, etc.
Such an abstract would describe the work done, and not the importance of the issue and the prospects, which are quite obvious.
- It would be possible to remove or shorten the first paragraph of the Introduction, as well as the left picture in figure 1, which can be found in any newspaper publication. Moreover, it is partially duplicated in Figure 4.
- In the introduction, it should be mentioned that ACT2 is far from the only target for the spike protein.
- Check for misprints – for example, p 7 line 178 – wascapable, etc.
Author Response
The article contains important, interesting, and extremely relevant information about the features of the receptor-binding domain (RBD) interaction of SARS-CoV-2 Spike glycoprotein and one of its targets, angiotensin-converting enzyme 2 (ACE2). The work combines well virtual methods that allow evaluating such interactions, with experimental approaches that confirm the assumptions made. The work is large and versatile. In my opinion, one could even break it down into several articles, in which one could focus on the individual aspects touched upon. For example, separately highlight the evolutionary aspects of the studied interaction. However, of course, it is possible to publish the article in the presented form without further revision after eliminating the following remarks.
We appreciate the recognition of the impact of our work by the reviewer.
- First of all, it is necessary to completely rewrite the abstract, removing obvious statements, and clearly and concisely present what was done in the work. Thus, the mention of the “addition of competitive inhibitors” leads the reader to assume that some special compounds have been investigated, although we are talking about protein fragments of the spike protein. So it should be written. Mention may be made of the lentivirus approach, evolutionary aspects, etc.
Such an abstract would describe the work done, and not the importance of the issue and the prospects, which are quite obvious.
Thank you for the constructive comment. We have reformatted the abstract to focus on the novel findings and importance of our work.
- It would be possible to remove or shorten the first paragraph of the Introduction, as well as the left picture in figure 1, which can be found in any newspaper publication. Moreover, it is partially duplicated in Figure 4.
We appreciate the suggestion to shorten the first paragraph of the introduction. However, we believe Figure 1 and 4 will help the general audience to better understand the main mechanism of our biosensor. Moreover, figure 4 shows a different mechanism and a spike-pseudotyped lentivirus as opposed to SARS-CoV2.
- In the introduction, it should be mentioned that ACE2 is far from the only target for the spike protein.
Thank you for bringing this point to our attention, we have made the corresponding changes to the introduction.
- Check for misprints – for example, p 7 line 178 – was capable, etc.
We have corrected the mentioned misprint along with any others we could find throughout our text.
